# Ferroelectric Nanodomain Engineering in Bulk Lithium Niobate Crystals in Ultrashort-Pulse Laser Nanopatterning Regime

**DOI:** 10.3390/nano12234147

**Published:** 2022-11-23

**Authors:** Sergey Kudryashov, Alexey Rupasov, Mikhail Kosobokov, Andrey Akhmatkhanov, George Krasin, Pavel Danilov, Boris Lisjikh, Anton Turygin, Evgeny Greshnyakov, Michael Kovalev, Artem Efimov, Vladimir Shur

**Affiliations:** 1Lebedev Physical Institute, 119991 Moscow, Russia; 2School of Natural Sciences and Mathematics, Ural Federal University, 620000 Ekaterinburg, Russia

**Keywords:** lithium niobate, ultrashort-pulse laser, bulk nanopatterning, ferroelectric nanodomains, engineering

## Abstract

Ferroelectric nanodomains were formed in bulk lithium niobate single crystals near nanostructured microtracks laser-inscribed by 1030-nm 0.3-ps ultrashort laser pulses at variable pulse energies in sub- and weakly filamentary laser nanopatterning regimes. The microtracks and related nanodomains were characterized by optical, scanning probe and confocal second-harmonic generation microscopy methods. The nanoscale material sub-structure in the microtracks was visualized in the sample cross-sections by atomic force microscopy (AFM), appearing weakly birefringent in polarimetric microscope images. The piezoresponce force microscopy (PFM) revealed sub-100 nm ferroelectric domains formed in the vicinity of the embedded microtrack seeds, indicating a promising opportunity to arrange nanodomains in the bulk ferroelectric crystal in on-demand positions. These findings open a new modality in direct laser writing technology, which is related to nanoscale writing of ferroelectric nanodomains and prospective three-dimensional micro-electrooptical and nanophotonic devices in nonlinear-optical ferroelectrics.

## 1. Introduction

During the last two decades, direct laser writing emerged as versatile tool in nano- and micro-fabrication of integrated photonic circuits, Bragg gratings, optical memory bits, microfluidic and optofluidic channels, phase and polarizing elements and devices in bulk dielectrics [1,2,3,4,5]. Inscription in a homogeneous dielectric medium requires a formation of a new high-contrast refractive-index interface, possible via densification in silica materials [6,7] and rarefaction in other dielectrics [8], boosting empty nanovoids [9] or drilling of hollow microchannels [10]. Meanwhile, rather recently—about one decade ago—a new fundamental high-NA (numerical aperture) laser inscription process (“bulk nanopatterning”, BN), relying on nanoplasmonic self-organization of birefringent grating arrays in bulk dielectrics as functional microbits, was matured for design and fabrication of innovative optical elements and devices. In the BN regime, multiple layers or continuous structure of birefringent character could be spatially arranged to manage flexibly light via combinations of polarization and interference operations [11,12,13,14]. Still, the promising BN modality should be explored regarding other broad classes of anisotropic ferroelectric and other functional crystalline dielectric materials.

Specifically, in ferroelectric crystals—e.g., lithium niobate (LN) and its doped modifications—laser-induced rarefaction [15] and BN regimes [16,17,18] were previously harnessed to manage non-linear optical circuitries, additionally utilizing electro-optical effect in electrical polarization domains [19]. The biaxial crystalline structure of this material resulted in rather unusual double linear focusing of ultrashort laser pulses in its bulk in the BN regime [16], while the related non-linear focusing and plasma phenomena, as well as post-pulse microstructural and atomistic transformations, were not explored yet. Hence, the overall picture of ultrashort-pulse laser BN inscription in bulk ferroelectric crystals, including LN, remains essentially incomplete, thus hindering progress in broad developing non-linear optical applications of these materials.

In this study, a bulk nanopatterning regime was employed for sub- and weakly filamentary laser inscription in LN single crystals by 0.65-NA focused, 1030 nm 0.3-ps laser pulses, and the resulting microstructures with their nanoscale sub-structure were comprehensively characterized by complementary optical, polarimetric, confocal second-harmonic generation, atomic force and piezoresponce force microscopy methods.

## 2. Materials and Methods

In these studies, we used for laser inscription congruent lithium niobate (CLN) z-cut and x-cut crystalline plates (Figure 1a), and a three-dimensional (3D) micro/nanostructuring laser workstation based on the femtosecond Yb-doped fiber laser system Satsuma (Amplitude Systemes, France), emitting pulses at the fundamental wavelength λ = 1030 nm (TEM_00_), pulsewidth τ ≈ 0.3 ps, variable pulse energy E = 0.1–10 μJ and repetition rate f = 0–500 kHz. The laser pulses were focused by a 0.65-NA micro-objective lens into the spot size w_0_ = 1.0 ± 0.1μm (1/e-intensity radius) at the depth of 50–60 µm inside the lithium niobate crystals (Figure 1b). The samples were mounted on a 3D motorized micro-positioner (Prior Scientific, UK) and scanned at the different pulse energies E = 112–660 nJ (fluence≈1.7–10 J/cm^2^, accounting for the 47% transmission coefficient of the optical path), repetition rate f = 10 kHz and translation speed of 400 μm/s, providing 60-µm wide, 3 mm long line arrays (series of lines separated by 3-micron distance) with the scan direction along the laser polarization. Here, in the anisotropic crystal the laser incidence and scanning directions were denoted below as Z/Y/, X/Z/, X/Y/ (incidence axis/scanning direction/). In the Z/Y case, the sample was irradiated via the Z-polar plane.

Threshold pulse energy for filamentation onset of the 1030-nm, 0.3-ps pulses in the CLN crystals were measured, using the same laser workstation, at higher f = 100 kHz and pulse energies E = 0.1–2.4 μJ (peak pulse power P = 0.3–8 MW). In the experiments, the laser pulses were focused by a 0.25-NA objective, while the images of the luminous filamentation channels were acquired sideways, using a 0.2-NA quartz/fluorite objective (LOMO, Russia) and a color CCD camera (Figure 1c). Laser pulses were introduced either via the z-cut polar plane, or via the y-cut plane, while the filament emission (transient recombination photoluminescence of electron-hole plasma) was acquired either via the y-cut plane, or via the z-cut plane. Moreover, the polarization azimuth of the 1030-nm, 0.3-ps laser pulses was varied to explore crystallographic effects.

The onset of filamentation was identified as an asymmetric elongation of the luminous channels (Figure 2a,b), depending on the laser pulse energy E (peak power P), reflecting the threshold-like appearance of the non-linear Kerr focus prior the Raleigh length of the focal region. Their comparison as a function of E or P allows (Figure 2c) to derive the critical power (threshold) energy for the filamentation onset (slightly different from the critical power for self-focusing). Specifically, such threshold powers P_th_ were measured in this way for the different plane cuts and polarizations: 0.84 ± 0.11 MW (X-cut, polarization along Z-axis), 0.77 ± 0.07 MW (X-cut, polarization along Y-axis), 0.54 ± 0.06 MW (Z-cut, polarization along X-axis) and 0.53 ± 0.06 MW (Z-cut, polarization along Y-axis). These threshold peak power values and the related pulse energies dictated us the inscription parameter window for the comparative sub- or weakly filamentary bulk nanopatterning at different sample arrangements regarding the laser incidence axis and polarization direction, which will be considered below.

Prior dicing of the CLN samples for microscopic characterization of the internal structure of laser-inscribed nanopatterns, non-invasive studies were performed by means of polarimetric and confocal Cherenkov-type second harmonic generation (CSHG) microscopy. Birefringence was characterized by a birefringence imaging system LCC7201B (Thorlabs, USA), exhibiting just minor variations of the retardance magnitude (λ/40-λ/20) versus laser pulse energy *E* ≤ 300 nJ (*P* ≤ 1 MW) (Figure 1d). The damage microtracks and related ferroelectric nanodomains were visualized inside the samples by means of CSHG microscopy, using a confocal microscope Ntegra Spectra (NT-MDT, Russia). An Yb fiber laser, operating at the 1064 nm wavelength, 5-MHz pulse repetition rate and average power of 50 mW of 2-ns pulses focused by 0.75-NA (40×), was used as a pumping source. The second harmonic signal at the 532 nm wavelength was acquired by a photo-multiplayer. A number of 3D CSHG images were reconstructed from the set of the 2D layer-by-layer scan images obtained with the 8-μm step in depth.

In order to reveal the ultrafine nanotopography of the buried fs-laser nanopatterned CLN, the inscribed linear horizontal arrays of vertical nanopatterns in the bulk CLN (Figure 1a,d) were saw-cut across the scan lines by an automated saw DAD-3220 (DISCO, Japan), using a Disco diamond blade disk Z09-SD3000-Y1-90 55x0.1 A2X40-L (DISCO, Japan). The cuts were consequently grinded by Al_2_O_3_ powders (grain sizes: 30, 9, and 3 μm) and polished by ≈25 nm colloidal SiO_2_ nanoparticles on the polishing machine PM5 (Logitech, UK) until optical surface quality reached the desired point. The uncovered cross-sectional topography was characterized in atomic force microscopy (AFM) and piezoresponse force microscopy (PFM) modes by means of an atomic force microscope NTEGRA Aura (NT-MDT, Russia), using Pt-coated NSC-18 probes (MikroMash, Russia, tip size—30 nm, first resonance frequency—400–500 kHz, stiffness coefficient—2.8 N/m) and 10-V, 20-kHz probing ac voltage. The typical 3D AFM/PFM images of cross-sectional relief and piezo-response are presented in Figure 3, Figure 4 and Figure 5.

## 3. Experimental Results and Discussion

### 3.1. AFM/PFM Visualization of Internal Nanopattern Structure

The acquired scanning probe (AFM, PFM) images in Figure 3, Figure 4 and Figure 5 indicate important trends in pulse-energy/peak power and crystallographic dependences of bulk nanopatterns in the z-cut and x-cut CLN samples in relation to the accompanying ferroelectric nanodomains. Specifically, in the Z/Y case in the sub-filamentary regime (E < 250 nJ, P < 0.84 MW) one can observe the nanodomains, spatially coinciding with the nanopatterned relief microtracks, but highly exceeding them in terms of their multi-micron length (Figure 3a–d). Only upon the filamentation onset at higher energy E = 300 nJ (P = 1 MW), the topography microtrack approaches the nanodomain length (Figure 3e,f). Overall, the gradual variation of the fs-laser pulse energy/peak power in the transition range between the sub- and weakly filamentary regimes enables in the sample the efficient up-scaling of the ferroelectric nanodomains in their length.

Similarly, all nanopatterned topography microtracks in the x-cut CLN sample in the sub- or weakly filamentary regime (E > 160 nJ, P > 0.53 MW) demonstrate their spatial coincidence with the nanodomains for the X/Z nanopatterns (Figure 4), where the isolated ferroelectric domains with sizes up to 400 nm are formed in the microtrack region (Figure 4d,h). Comparing to the Z/Y nanopatterns inscribed in z-cut CLN (Figure 3), in this nanopatterning regime the microtracks exhibit much more fine and regular structure (Figure 4a,b), appearing merged at the higher energy of 300 nJ (Figure 4 e,f). This is obviously in agreement with the lower threshold power for filamentation in this regime. Moreover, in this regime the induced nanodomains appear visibly smaller, once grown along the Z-axis into the figure plane.

This is also the case for the X/Y nanopatterns inscribed in the x-cut CLN in the weakly filamentary regime at the same energies of 150 nJ (P = 0.5 MW) and 300 nJ (P = 1 MW) in Figure 5. In this case, the nanopatterns appear less indistinctly, but sill periodically with the deeply subwavelength periods ~100–200 nm (Figure 5c,d). According to the laser arrangement and CLN sample orientation, the isolated ferroelectric nanodomains were elongated along the polar axis (Z-axis), demonstrating the needle-like shape for the forward growth (Figure 5f).

In addition, CSHG cross-sectional images of the ferroelectric nanodomains inscribed along with the microtracks in the x-cut and z-cut CLN samples in the weakly filamentary regime, provide the unambiguous indication of the present ferroelectric structures [17,18,20] (Figure 6). Similar to the AFM/PFM images in Figure 3, Figure 4 and Figure 5, the microtracks emerged in the CSHG images in Figure 6 with the longer tracks at the higher pulse energies. Importantly, in the X/Z case each even microtrack appears thinner/shorter in Figure 6b than the odd one, indicating the recently demonstrated non-reciprocal character of the laser inscription along the polar axis in opposite directions [21].

Finally, the filamentation effect of ultrashort laser pulses in LN samples was previously supposed, but not analyzed in [16,17,18,19,20,21]. Here, through our explicit visualization measurements of 1030-nm, 0.3-ps laser threshold pulse energies/peak laser powers for the filamentation onset in CLN, we provide the straightforward demonstration of its relationship with the structural/topographic damage and ferroelectric nanodomain engineering in CLN.

### 3.2. Ultrashort-Pulse Laser Control of Nanopattern and Ferroelectric Nanodomain Lengths

The relationship between the topography nanopattern and ferroelectric nanodomain lengths in the explored sub- and weakly filamentary regimes, which are presented in Figure 3, Figure 4, Figure 5 and Figure 6 for the different crystallographic and laser polarization orientations, is overviewed below in Figure 7. Here, the results of the optical microscopy (OM) and AFM measurements of the laser-inscribed nanopattern lengths at *P* ≤ 1 MW (Figure 2) are compared with the PFM-envisioned ferroelectric nanodomain lengths, exhibiting, in general, their good correspondence versus the laser pulse energy or peak power.

Meanwhile, importantly, one can see that in some CLN orientations and laser inscription arrangements the diminished laser pulse energy enables nanodomain downscaling till ~1-μm dimensions (Z/Y nanopatterning regime, Figure 7a), or upscaling such nanodomain lengths at higher energies along the laser propagation path and nanopatterning track (Figure 3). Moreover, in this regime, the onset of fs-laser filamentation induces the rapid growth of the nanopatterned region observed by OM and AFM, until the corresponding nanodomain lengths visualized by PFM, along the polar axis Z (Figure 7a) or the non-polar X-axis (Figure 7c). Surprisingly, scanning in the Z-direction, the laser beam incident in the Z-direction provides no considerable difference between the nanopattern and nanodomain lengths (Figure 7b). Overall, these findings open the new experimental opportunities in managing and exploring the relationship between topographic nanopatterns and ferroelectric nanodomains in lithium niobate crystals.

### 3.3. Inscription Mechanism of Ferroelectric Nanodomains in Bulk CLN

The ferroelectric domain structures obtained in the vicinity of the fs-laser inscribed nanopatterns in this work can be explained by one of three alternative mechanisms presented below.

According to the first mechanism, polarization reversal could occur under the action of a depolarization field produced by bound charges, which are localized at the boundaries of the damaged nanopatterned regions. Here, we suggest that our fs-laser pulses induce amorphization of the bulk CLN material and its transition to a nonpolar state. As a result, the bound charges appear at the boundary between the virgin and damaged crystal. The depolarization field produced by the bound charges leads to the appearance of nanodomains at the boundary and their growth into the virgin material. Such mechanism has been discussed for relaxor ferroelectrics to explain formation of a nanodomain structure after heating of a single-domain crystal from a ferroelectric to a relaxor phase, consisting of isolated nonpolar regions within a ferroelectric matrix [22,23].

According to the second mechanism, polarization reversal could be induced by a pyroelectric field appeared during local CLN cooling after each heating fs-laser pulse. It is known that surface irradiation of a single-domain z-cut CLN sample by a pulsed infrared laser leads to formation of submicron-width stripe domains within the irradiated area [24,25]. The proposed mechanism was previously confirmed by computer simulations of a temporal evolution of a pyroelectric field, exceeding in its magnitude the threshold value [24,25].

According to the third mechanism, the polarization reversal could occur due to the action of a thermoelectric field, induced by a temperature gradient in the irradiated area due to its inhomogeneous sample heating. This mechanism was considered recently in the analysis of ferroelectric domain structures, emerging in MgO-doped CLN crystals as a result of fs-laser irradiation [21].

At this time, according to the current status of our experimental studies each of the abovementioned mechanisms could explain the ferroelectric domains formation in our study and further time-resolved in situ ultrafast Raman studies, e.g., similar to those reported in [26], are in our plans be done for distinguishing between the electronic and thermal effects in the polarization reversal and revealing the most relevant mechanism.

## 4. Conclusions

In this study, 1030-nm, 0.3-ps laser pulses were used to inscribe bulk structural/topographic nanopatterns and related ferroelectric nanodomains in lithium niobate single crystals in sub- and weakly filamentary propagation/focusing regimes. The characterization of the inscribed nanostructures by a set of optical, scanning probe (atomic and piezoresponce force) and confocal second-harmonic generation microscopy methods enabled to reveal the possibility of their up/down-scaling as a function of laser pulse energy (peak power), depending on the crystallographic orientations (plane cuts) and arrangement of the crystals during the laser inscription. Our findings revealed sub-100 nm ferroelectric domains formed in the vicinity of the microtrack seeds, indicating a promising opportunity to arrange nanodomains in on-demand positions inside the material, as a new modality in direct laser writing technology for ferroelectric nanodomain engineering in bulk lithium niobate.

## Figures and Tables

**Figure 1 nanomaterials-12-04147-f001:**
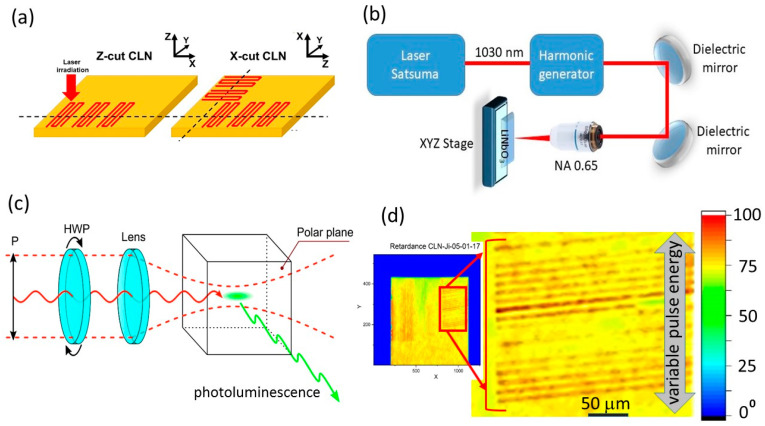
(**a**) CLN arrangement, crystal axis orientations and inscription patterns (laser incidence axis/scan directions: Z/Y, X/Z, X/Y), with their cut cross-sections indicated by the dark dashed lines. (**b**) Layout of laser workstation. (**c**) Side-view microscopic photoluminescence (PL) visualization of laser filaments and critical power acquisition in CLN. (**d**) Polarimetric microscopic images of laser tracks, exhibiting different phase shifts [color scale, ^0^], depending on laser pulse energy.

**Figure 2 nanomaterials-12-04147-f002:**
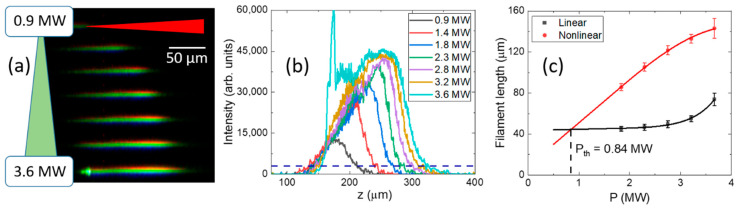
(**a**) Side-view microscopic PL visualization of fs-laser filaments as a function of peak laser power. (**b**) Longitudinal profiles of the asymmetric PL filamentary channels (laser comes from the right side) as a function of peak laser power. (**c**) Comparison of power-dependent PL region (non-linear focusing) and Rayleigh (linear focusing) lengths.

**Figure 3 nanomaterials-12-04147-f003:**
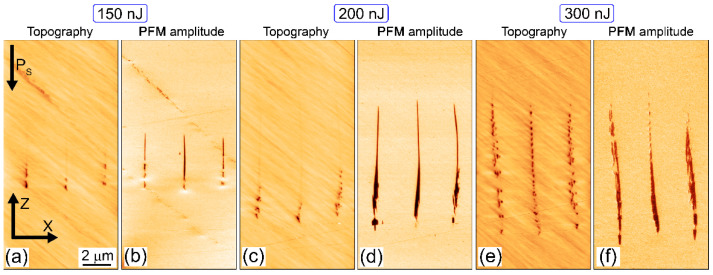
Topography/relief depth (**a**,**c**,**e**) and PFM amplitude (**b**,**d**,**f**) scan images of Z/Y nanopatterns inscribed in z-cut CLN at different energies:150 (**a**,**b**), 200 (**c**,**d**) and 300 (**e**,**f**) nJ. The laser pulses come from the top.

**Figure 4 nanomaterials-12-04147-f004:**
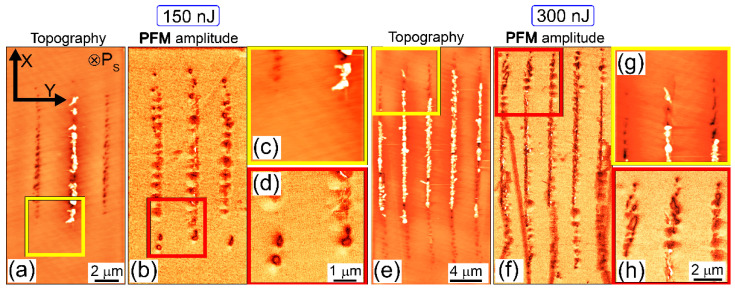
Topography/relief depth (**a**,**c**,**e**,**g**) and PFM amplitude (**b**,**d**,**f**,**h**) scan images of X/Z nanopatterns inscribed in x-cut CLN at different energies of 150 (**a**–**d**) and 300 (**e**–**h**) nJ. The laser pulses come from the top.

**Figure 5 nanomaterials-12-04147-f005:**
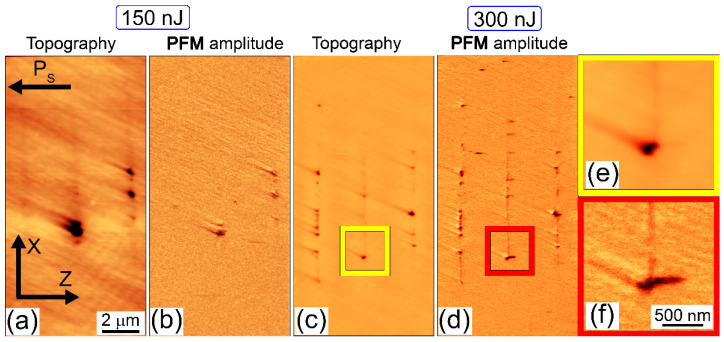
Topography/relief depth (**a**,**c**,**e**) and PFM amplitude (**b**,**d**,**f**) scan images of X/Y nanopatterns inscribed in x-cut CLN at different energies of 150 (**a**–**c**) and 300 (**d**–**f**) nJ. The laser pulses come from the top.

**Figure 6 nanomaterials-12-04147-f006:**
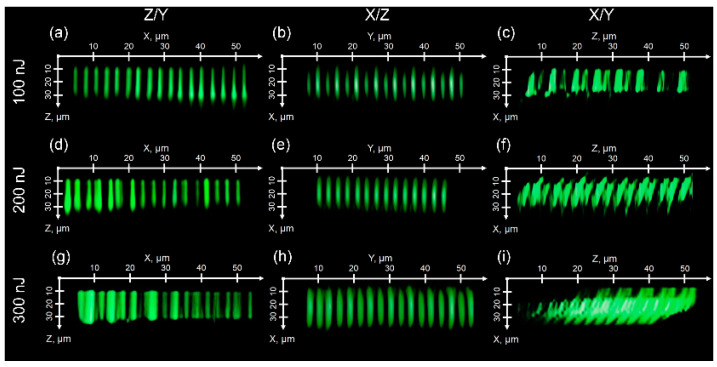
CSHG cross-sectional images of microtracks with nanodomains inscribed at different energies of 100 (**a**–**c**), 200 (**d**–**f**) and 300 (**g**–**i**). nJ inscribed in z-cut CLN (**a**,**d**,**g**—Z/Y nanopatterns), (**b**,**e**,**h**—X/Z nanopatterns) and (**c**,**f**,**i**—X/Y nanopatterns) nanopatterns inscribed in x-cut CLN samples.

**Figure 7 nanomaterials-12-04147-f007:**
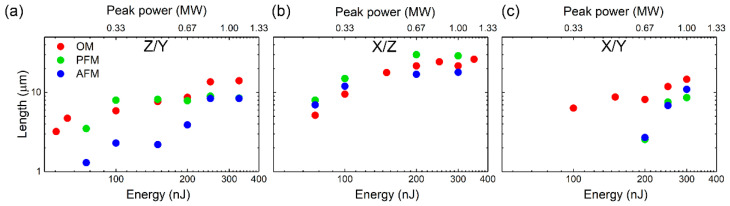
AFM (blue circles) and optically measured (optical microscopy, OM, red circles) lengths of nanopatterned microtracks, in comparison to PFM-envisioned nanodomain lengths (green circles) versus laser pulse energy (bottom axis) and peak pulse power (top axis) for Z/Y (**a**), X/Z (**b**) and X/Y (**c**) orientations. Error bars ≈1 μm. See the same scales for the ordinate axes.

## Data Availability

The data supporting the reported results can be obtained from the authors upon a reasonable request.

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
