# Peer review of "Ferroelectric Nanodomain Engineering in Bulk Lithium Niobate Crystals in Ultrashort-Pulse Laser Nanopatterning Regime"

_nanomaterials, 2022, doi:10.3390/nano12234147_

Round 1

Reviewer 1 Report

The manuscript is a well written and presented technical paper. I have no issues to it.

"The  manuscript deals with the possibility to form nanodomains in lithium niobate in proximity to the damage induced in the bulk material by irradiation with an ultrashot infrared laser. This technique could open up the possibility to directly use lasers to micropatterning of materials. The authors accurately analyze by means of several techniques the outcome of the micropatterning process and hypothesize three possible microscopic mechanisms that might be responsible for the results they obtained. The technique set up by the authors appears to be quite innovative. The manuscript is an accurate piece of material manufacturing techniques. It is well written and organized and the references are adequate. In my opinion it is appropriate for the journal focus and should be accepted as is."

Author Response

Replies to the Reviewers (in bold)

Reviewer 1

"The  manuscript deals with the possibility to form nanodomains in lithium niobate in proximity to the damage induced in the bulk material by irradiation with an ultrashot infrared laser. This technique could open up the possibility to directly use lasers to micropatterning of materials. The authors accurately analyze by means of several techniques the outcome of the micropatterning process and hypothesize three possible microscopic mechanisms that might be responsible for the results they obtained. The technique set up by the authors appears to be quite innovative. The manuscript is an accurate piece of material manufacturing techniques. It is well written and organized and the references are adequate. In my opinion it is appropriate for the journal focus and should be accepted as is."

We appreciate the high evaluation of out work.

Reviewer 2

 The authors present “Ferroelectric nanodomain engineering in bulk lithium niobate crystals in ultrashort-pulse laser nanopatterning regime”, which systematically described the materials and methods, experimental results and discussion. The manuscript itself is interesting, scientific and useful investigation for the integrated photonic circuits, optical memory bits, microfluidic and optofluidic channels, phase and polarizing elements and devices. However, before this manuscript is accepted there are a lot of problems need to be addressed:

  1. The English needs to be further polished in the revised manuscript (such as: 1030-nm 0.3-ps, 60-mm, 3-mm, 2-ns, 8-mm, 25-nm, 532-nm, 10-v, 20-kHz,…. etc).

According the 5-year US teaching/research experience of the corresponding author, this is a standard short form of composite adjectives in concise scientific writing style: for example, 25-mm length = length of 25 mm.

  1. It is recommended that authors need to rewrite the Keywords. Done
  2. Write the complete meaning of the abbreviation when it first appears in the manuscript. E.g: TEM00, LOMO, PL. Fixed
  3. In Figures 4 and 5, the authors need to explain their results in more detail. Provided
  4. In Figure 6, CSHG cross-sectional images with pulse energy of 30 nJ should be added. Added for 300 nJ
  5. In the content of the manuscript, it is necessary to explain and compare the measurement results of OM, PFM and AFM in Figure 7. Explanation and comparison of the data in Fig.7 was provided.
  6. The relation between the research results of this paper and the three mechanisms mentioned in content 3.3 is not clear enough. Additional analysis of our results in the context of the three hypothetical mechanisms was provided.

I recommend that the paper will be published in nanomaterials with major revision.

The major revision of the manuscript was performed according to the Reviewer’s suggestions.

Reviewer 2 Report

  The authors present “Ferroelectric nanodomain engineering in bulk lithium niobate crystals in ultrashort-pulse laser nanopatterning regime”, which systematically described the materials and methods, experimental results and discussion. The manuscript itself is interesting, scientific and useful investigation for the integrated photonic circuits, optical memory bits, microfluidic and optofluidic channels, phase and polarizing elements and devices. However, before this manuscript is accepted there are a lot of problems need to be addressed:

 1. The English needs to be further polished in the revised manuscript (such as: 1030-nm 0.3-ps, 60-mm, 3-mm, 2-ns, 8-mm, 25-nm, 532-nm, 10-v, 20-kHz,…. etc).

2.  It is recommended that authors need to rewrite the Keywords.

3. Write the complete meaning of the abbreviation when it first appears in the manuscript. E.g: TEM00, LOMO, PL.

4.  In Figures 4 and 5, the authors need to explain their results in more detail.

5.  In Figure 6, CSHG cross-sectional images with pulse energy of 30 nJ should be added.

6. In the content of the manuscript, it is necessary to explain and compare the measurement results of OM, PFM and AFM in Figure 7.

7.  The relation between the research results of this paper and the three mechanisms mentioned in content 3.3 is not clear enough.

I recommend that the paper will be published in nanomaterials with major revision.

Author Response

(The authors gave the same response as above.)

Round 2

Reviewer 2 Report

The authors had made appropriate revised based on the comments of the previous review. I recommend that the paper can be published in nanomaterials.